

# Gene synthesis design: a pythonic approach

Yunzhuo Hu[1,2], Danni Pan[1,2], Fei Xu[1,2], Bifang Huang[3], Xuanyang Chen[1,2] and Shiqiang Lin[1,3]

[1] Agricultural Product Quality Institute, Fujian Agriculture and Forestry University, Fuzhou, Fujian, China
[2] College of Agronomy, Fujian Agriculture and Forestry University, Fuzhou, Fujian, China
[3] College of Life Science, Fujian Agriculture and Forestry University, Fuzhou, Fujian, China

## ABSTRACT

Researchers often need to synthesize genes of interest in this era of synthetic biology. Gene synthesis by PCR assembly of multiple DNA fragments is a quick and economical method that is widely applied. Up to now, there have been a few software solutions for designing fragments in gene synthesis. However, some of these software solutions use programming languages that are not popular now, other software products are commercial or require users to visit servers. In this study, we propose a Python program to design DNA fragments for gene synthesis. The algorithm is designed to meet the experimental needs. Also, the source code with detailed annotation is freely available for all users. Furthermore, the feasibility of the algorithm and the program is validated by experiments. Our program can be useful for the design of gene synthesis in the labs and help the study of gene structure and function.

# INTRODUCTION

Gene cloning is necessary for the investigation of protein structure and function. Normally, the target gene is amplified with PCR from the template. The prokaryotic genes can be obtained by using the genome as a template, while in the case of a eukaryotic gene, researchers extract RNA, which is then reverse-transcribed to cDNA that serves as the template for target gene amplification. However, under some circumstances, the cells harboring the gene of concern are not available so there is no template for PCR amplification. Researchers will then need to synthesize the target gene for functional study. Sometimes, codon optimization is performed to improve target gene expression, which is frequently involved in lots of codon substitution. The wild-type gene cannot act as the template to get the optimized gene and this is virtually the same as having no template at hand. There is at least one more situation. Nowadays, researchers are expanding the range of study from naturally existing genes to include genes whose sequences are generated by artificial intelligence (*Madani et al., 2023*). These genes have to be synthesized before their structure and function can be experimentally investigated (*Cox & Blazeck, 2022*; *Korendovych & De Grado, 2020*; *Singh et al., 2018*).

Corresponding authors
Xuanyang Chen, cxy@fafu.edu.cn
Shiqiang Lin, linshiqiang@fafu.edu.cn

For the target gene to be synthesized, if it is short, less than 100 bps, for example, we can directly synthesize the sense strand and the antisense strand, anneal them, and ligate to the T-vector. If the target gene is long, then DNA fragments with overlaps are synthesized and subsequently linearly assembled into the full-length gene with PCR. During the linear assembly, the consecutive fragments contain overlap regions that enable bridging to get the joined sequence. As the PCR reaction goes, the assembled sequence becomes longer and longer, till the full-length gene is reached. The nature of the assembly process is linear, therefore, there is only a small quantity of full-length genes. However, the full-length gene can be used as the template for exponential amplification with gene primers. Then we will get an enough amount of target gene for gel recycling and T-vector ligation (*Prodromou & Pearl, 1992*; *Stemmer et al., 1995*; *Xiong et al., 2004*).

Currently, several software solutions for gene synthesis design have been reported such as DNAWorks (*Hoover & Lubkowski, 2002*), Gene2Oligo (*Rouillard et al., 2004*), GeMS (*Jayaraj, Reid & Santi, 2005*), Assembly PCR oligo maker (*Rydzanicz, Zhao & Johnson, 2005*), GeneDesigner (*Villalobos et al., 2006*), GeneDesign 3.0 (*Richardson et al., 2010*), *etc.,* which have greatly promoted the development of gene synthesis. In the age of artificial intelligence, Python is the most popular programming language among the scientific community. Therefore, we use Python to write the program for gene synthesis design. Two methods of implementation are introduced here (Fig. 1). The method (A) is simple and intuitive, of which the principle can be easily understood and the experimental design is also convenient. Each PCR cycle joins one more DNA fragment, and we can calculate the minimal PCR cycle number for full-length gene assembly according to this reaction characteristic. The method (B) joins the fragments to the full-length gene by overlap bridging as in the method (A). The difference is that in the method (B) pairs of neighboring fragments are allowed to merge to longer sequences in each PCR cycle till the full-length gene is assembled, while in the method (A) the full-length gene must be assembled from the last fragment to the first fragment one by one.

According to the methods of (A) and (B) for full-length gene assembly, we design the algorithm and program and perform experimental validation, aiming to provide researchers with an elegant, free, and open-source method of gene synthesis design that assists the structural and functional study of genes.

## MATERIALS AND METHODS
### Computer hardware and software
The computer hardware is MacBook Air (M1, 2020) and the operating system is macOS Monterey 12.4. The script running needs Python3.10 (http://www.python.org), biopython 1.79 (*Cock et al., 2009*), and matplotlib 3.6.3 (*Hunter, 2007*). The IDLE (Integrated Development and Learning Environment) of Python3.10 is used to write and edit the script. Our script for gene synthesis 'gene_synthesis.py' (https://github.com/shiqiang-lin/gene-synthesis), example gene sequence file 'beta_original.fasta' (https://github.com/shiqiang-lin/gene-synthesis), 'beta_optimized.fasta' (https://github.com/shiqiang-lin/gene-synthesis) are available at https://github.com/shiqiang-lin/gene-synthesis.

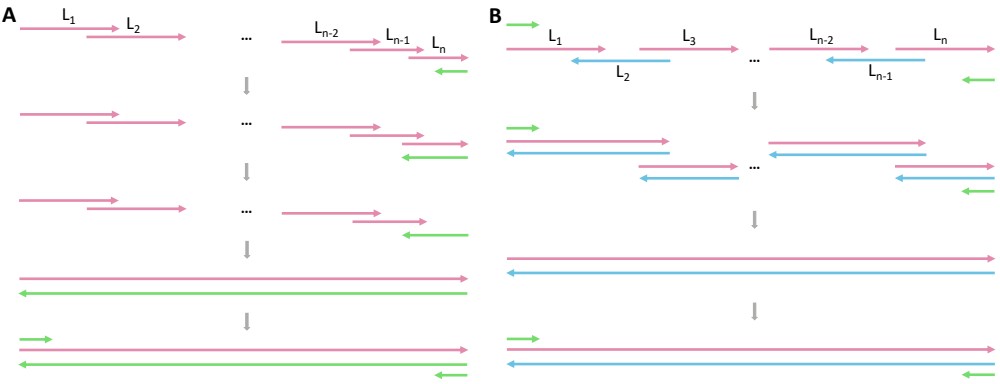

**Figure 1** **Principle of gene synthesis *via* PCR.** (A) Using all sense fragments. (B) Using alternating sense and antisense fragments. $L_1$, $L_2$, …, $L_n$ (short pink arrows) indicate the DNA fragments, with each pair of neighboring fragments sharing an overlap region. The primers for the full-length gene are represented by green arrows. In (A), the PCR for gene assembly begins with the annealing of antisense primer to $L_n$. Then the reverse complement sequence of $L_n$ is generated with DNA polymerase. During the next PCR cycle, the reverse complement of $L_n$ bridges the $L_{n-1}$ in the overlap region, and $L_{n-1}$-$L_n$ is obtained with DNA polymerase, and so on. As the PCR goes, the full-length sense sequence and the full-length antisense sequence are assembled, which can be used as the template for the rest of PCR cycles. In (B), the blue arrows show the antisense sequences. Each pair of neighboring fragments can join to a longer fragment with overlap bridging during PCR assembly.

## Algorithm

The algorithm of the program is shown in Fig. 2 and listed as follows. (1) Set the parameters, set_fragment_len for DNA fragment length and Tm_set for the range of temperatures to be screened, and get DNA fragment $L_1$ with a length of set_fragment_len from the 5′ end of the gene to be synthesized; (2) select a certain length of sequence from the 3′ end of $L_1$, with its Tm the closest to the Tm_set, and this sequence is the overlap between $L_1$ and $L_2$. The remaining gene sequence after cutting off $L_1$ is defined as remaining_gene_sequence_1; (3) see the length of the overlap between $L_1$ and $L_2$ + remaining_gene_sequence_1. If the length is equal to or greater than set_fragment_len, then the $L_2$ can be obtained from the overlap of $L_1$ and $L_2$ and a certain length of sequence in the 5′ end of remaining_gene_sequence_1. The length of $L_2$ is set_fragment_len. In this way, we can get $L_3$, $L_4$, …, till $L_{n-1}$. If the sum of overlap between $L_{n-1}$ and $L_n$ and remaining_gene_sequence_n-1 is less than set_fragment_len, then see if the remaining_gene_sequence_n-1 is less than 20bps. If it is, then take 20bps from the 3′ end of $L_{n-1}$ and add it to the 5′ end of remaining_gene_sequence_n-1, and redo the previous step to get the new overlap sequence and the $L_n$. If it is not, then $L_n$ can be obtained by joining the overlap between the $L_{n-1}$ and $L_n$ and the remaining_gene_sequence_n-1.

For the overlap region, the Tm of the sense sequence is equal to the Tm of the antisense sequence. Therefore, our algorithm here applies to both (A) and (B) (Figs. 1A and 1B). Actually, in the presence of primers gene_5 and gene_3, any DNA fragments in (A) or (B) can be changed to their reverse complement sequences and we are still able to get the full-length gene.

**Figure 2** **A schematic diagram of the algorithm.** The longer black arrow represents the target gene sequence (gene_sequence). $L_1$, $L_2$, …, $L_{n-1}$, and $L_n$ are the program-generated fragments for target gene assembly. For the neighboring fragments, there is an overlap between the 3′ end of the previous fragment and the 5′ end of the next fragment. For example, the blue part is the overlap between $L_1$ and $L_2$. The remaining_gene_sequence_1 shows the rest of the target gene sequence after truncating $L_1$, and remaining_gene_sequence_2 shows the rest of remaining_gene_sequence_1 after truncating $L_2$ (precisely, $L_2$ minus overlap between $L_1$ and $L_2$), and so on. The set_fragment_len is the program-set parameter for defining the length of DNA fragments for gene synthesis.

## Running method

Here we use the example file 'beta_original.fasta', which is the original gene sequence before sequence optimization, to show the running process of our script. (1) Perform gene sequence optimization with the Codon Optimization Tool (https://sg.idtdna.com/pages/tools/codon-optimization-tool) from Integrated DNA Technologies, Inc. (IA, USA), and obtain the optimized gene sequence 'beta_optimized.fasta'. This step can not only improve gene expression but also simplify the DNA structure to help PCR synthesis of the target gene. Other commercial companies provide sequence optimization services, for example, the GenSmart Codon Optimization (https://www.genscript.com/gensmart-free-gene-codon-optimization.html) of GenScript (NJ, USA); (2) Make a new directory on the Desktop and copy 'gene_synthesis.py' and 'beta_optimized.fasta' to the directory; (3) Open Terminal and cd to the directory in step (2), input the following command, and press 'Enter' to run the script;

python3.10 gene_synthesis.py beta_optimized.fasta

(4) In one or two seconds, a scatter diagram appears (Fig. 3), showing the Tm of each fragment.

At this time, you can move the mouse to the dot in the diagram to see the exact Tm value. The horizontal line in the diagram shows the Tm_mean for all overlaps. You can save the diagram or just close the window without saving the diagram. After closing the scatter diagram, a new folder named 'gene_synthesis_results' appears in the directory of step (2). Within the folder, there are three files, which are 'all_sense_fragments.txt', 'gene_primers.txt', and 'sense_antisense_fragments.txt'. The file 'all_sense_fragments.txt' stores the designed fragments (all sense sequences), overlap sequence, Tm, Tm-Tm_mean, fragment_len, overlap_len, and start_pos-end_pos. The file 'sense_antisense_fragments.txt' stores the designed fragments (with alternating sense sequences and antisense sequence), overlap sequence (using sense sequence), Tm, Tm-Tm_mean, fragment_len, overlap_len,

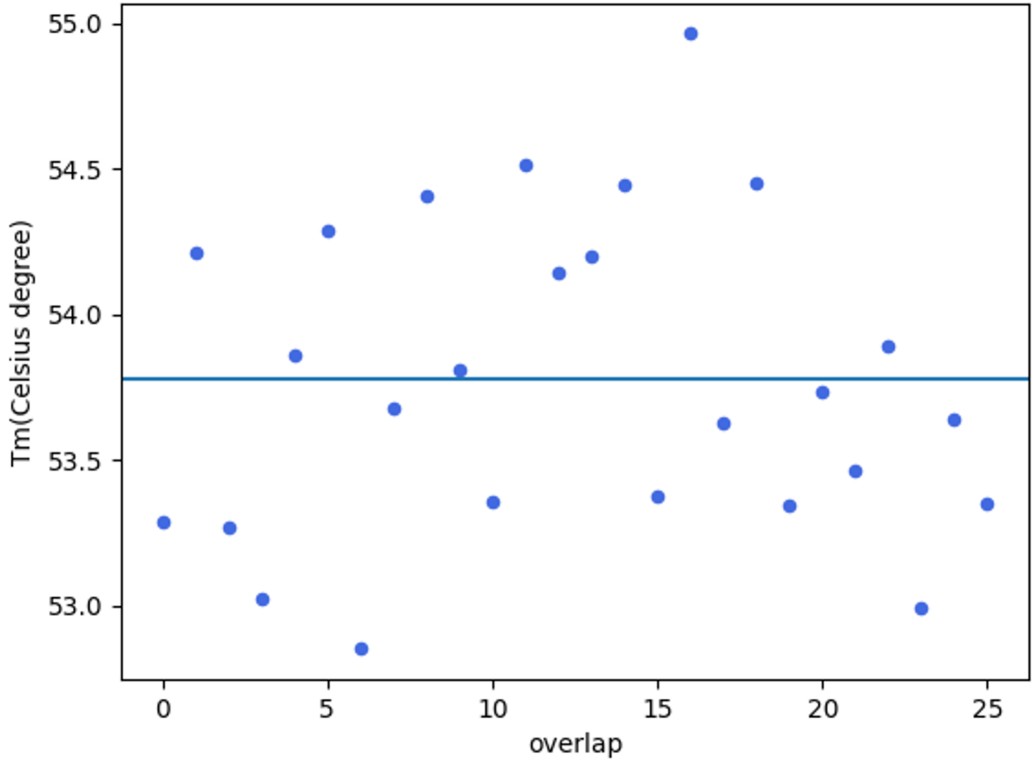

**Figure 3** **Tm values of overlaps.**

and start_pos-end_pos. The file 'gene_primers.txt' lists the gene_primer sequence, Tm, and Tm-Tm_mean.

## Experimental validation

The output files of the program contain the method (A) and method (B) in Fig. 1; therefore, we conducted experimental verification for both methods. The length of the target gene is over 1,000bps. The output text files for method (A) and method (B) both contain 27 DNA fragments. We define three blocks, each including 9 DNA fragments. Because the block definition is the same for method (A) and method (B), we are able to compare the difference between the two experimental processes. The detailed verification process is described as follows.

The file 'beta_optimized.fasta' is used as an example of the program running. The DNA sequences that were sent for commercial synthesis include fragments, sense and antisense overlap sequences (used as primers to amplify blocks, *i.e.,* block primers), and gene primers, which were collected and stored in 'all_synthsized_sequences.txt' (https://github.com/shiqiang-lin/gene-synthesis). All sequences were sent to Sangon Biotech Co., Ltd. (Shanghai, China) for synthesis. During the process of gene synthesis with PCR, there is a limit to the number of DNA fragments that can be successfully assembled. In theory, it is more difficult in the situation of more DNA fragments. It has been reported that six or seven fragments can be easily assembled with PCR

(*Xiong et al., 2004*). In this study, we attempted to join nine fragments for each block (https://github.com/shiqiang-lin/gene-synthesis). Each block was amplified with the nine DNA fragments as template and the block primers, using Pyrobest DNA Polymerase (Takara, catalog NO. R005A). The PCR system (50 ul) contained 5 ul 10×Pyrobest Buffer II (Mg$^{2+}$ plus, 10 mM), 2 ul 10 uM sense block primer, 2 ul 10 uM antisense block primer, 4 ul dNTP mixture (2.5 mM each), nine DNA fragments (10 uM for each fragment), 0.25 ul Pyrobest DNA Polymerase (5 U/ul), and ddw. The PCR program was 94 °C for 1min; 35 cycles of 98 °C for 2 s, 54 °C for 30 s, 72 °C for 25 s; 16 °C forever.

After three blocks were obtained with PCR reactions, 1.5% agarose gel was used to isolate the target DNA bands (theoretical length in bps in https://github.com/shiqiang-lin/gene-synthesis). The target bands were cut and recycled with Gel Extraction Kit (Sangon, catalog NO. B58131-0100), using 30ul Elution Buffer to elute target DNA. The DNA concentration is determined with NanoDrop (Thermo Fisher Scientific Inc., MA, USA) (https://github.com/shiqiang-lin/gene-synthesis). Then the three blocks were used as the template to assemble and amplify the full-length gene. The PCR system (50 ul) contained 5 ul 10×Pyrobest Buffer II (Mg$^{2+}$ plus, 10 mM), 2 ul 10 uM sense gene primer, 2 ul 10 uM antisense gene primer, 4 ul dNTP mixture (2.5 mM each), three blocks (50 ng for each block), 0.25 ul Pyrobest DNA Polymerase (5 U/ul), and ddw. The PCR program was 94 °C for 1 min; 35 cycles of 98 °C for 2s, 54 °C for 30 s, 72 °C for 1 min 10 s; 16 °C forever. The target gene band was gel recycled using Gel Extraction Kit (Sangon, catalog NO. B58131-0100), ligated to T-vector using Zero TOPO-Blunt Cloning Kit (Sangon, catalog NO. B522216-0020), and sent for sequencing verification (https://github.com/shiqiang-lin/gene-synthesis).

## RESULTS

### Program running

For a smooth process of DNA fragment assembly, it is often necessary to perform sequence optimization for the target gene. Sequence optimization helps to improve gene expression, eliminate or elevate DNA secondary structure, and adjust regions of too high or too low GC content. Here, we utilize the free service from commercial companies to get the optimized gene sequence. In practice, the fragment length will have to conform to the synthesis quality and cost control. Besides, all the overlaps between the consecutive fragments and the gene primers have approximate Tm values, to facilitate the fragment joining. Based on these considerations, we run the program using the example gene sequence file as input, and the results are shown in Fig. 4.

There are seven columns in Fig. 4A, including fragment sequence, overlap sequence, Tm of overlap sequence, Tm-Tm_mean, fragment_len, overlap_len, and start_pos-end_pos. It can be seen that all the fragments are sense sequences. In Fig. 4B, the first column shows fragments in alternating sense antisense sequences. The odd fragments are sense and even fragments are antisense. The rest six columns are the same as those in Fig. 4A. The primer, Tm value, and Tm-Tm_mean for amplification of the full-length gene are displayed in Fig. 4C. It is shown that the length of each fragment is 59 bps, the

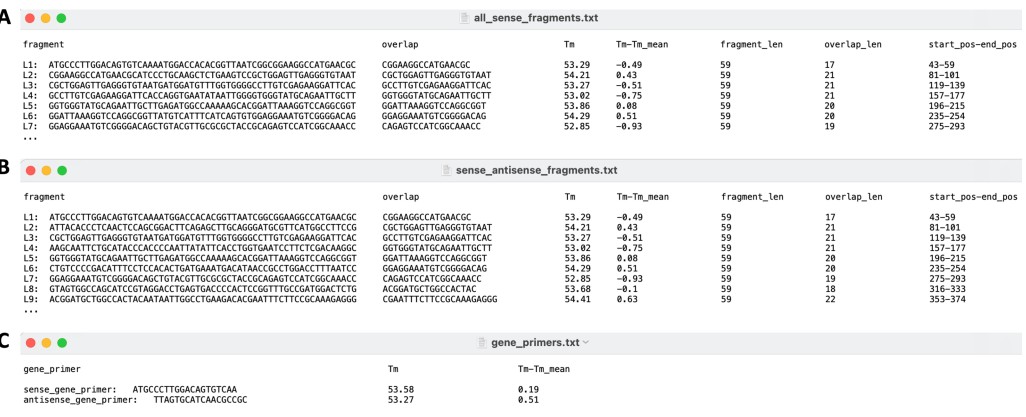

**Figure 4** **The output files (partial) of the program running.**

same as the program definition (which can be adjusted for quality and cost control). All the fragments have close Tm values, with Tm-Tm_mean 0.93 the highest (less than 1) (https://github.com/shiqiang-lin/gene-synthesis). The above results show that the Tm of overlap for each fragment accords with the program design. In the case of gene primers, Tm-Tm_mean is 0.51, far less than 1, consistent with the program design as well. It should be noted that the overlaps do not have a constant length. The reason is that the program screens overlap in length to keep all the Tm values close. These results lay a solid foundation for the experiment of synthesizing the target gene with PCR.

## Experimental validation

We synthesized all the needed DNA sequences for method (A) and method (B) according to the program output, which includes fragments, block primers, and gene primers. We utilized the idea of 'divide and conquer' to address the issue of gene synthesis. The full-length gene was divided into several blocks, with each block made up of multiple commercially synthesized DNA fragments. During experiments, PCR assembly and amplification is performed for each block using the constituting fragments as the template, and for the full-length gene using the blocks as the template. The experimental result of method (A) is shown in Fig. 5.

It can be seen from lanes 1–3 that all the blocks (https://github.com/shiqiang-lin/gene-synthesis) were PCR assembled and amplified with their corresponding fragments as templates. The target bands are bright and clear-cut, which are convenient for cutting gels and recycling DNA. The concentrations of three blocks all exceeded 30 ng/ul, and these were used as the template to assemble and amplify the full-length gene. The band in lane 4 shows that the full-length gene was PCR synthesized successfully, and the quality and concentration of three blocks were eligible for overlap bridging. It is supposed that the close Tm values for all overlaps and gene primers played an important role during the processes of block assembly with fragments and full-length gene assembly from blocks, which is realized by our algorithm design and Python code. Having seen the experimental
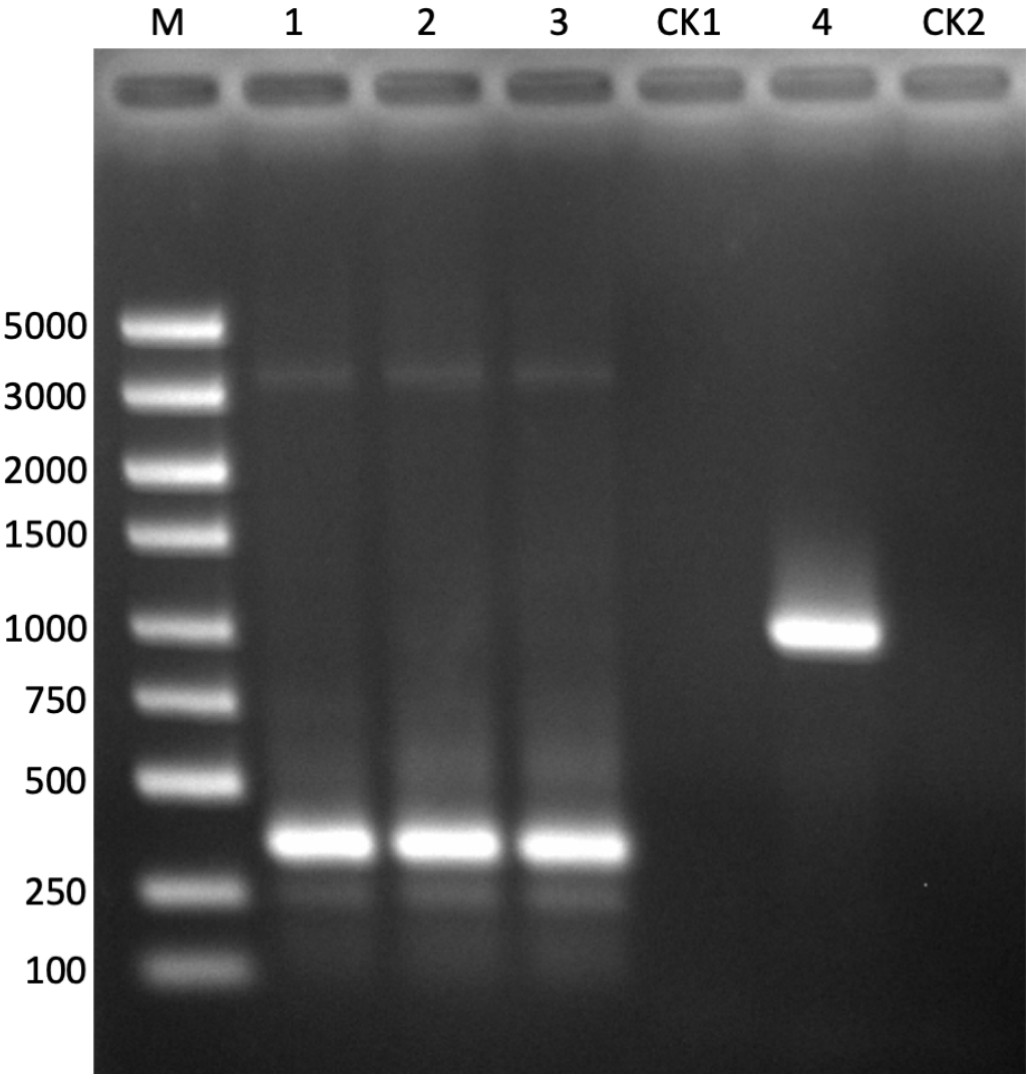

**Figure 5** **The validation of method (A).** M: DNA marker; lane 1: block1; lane 2: block2; lane 3: block3; CK1: negative control for lanes 1–3 with ddw as the template; lane 4: blocks 1–3; CK2: negative control for lane 4 with ddw as the template.

result of method (A), we now turn to the experimental result of method (B), which is shown in Fig. 6.

Lanes 1–3 show that all the blocks (https://github.com/shiqiang-lin/gene-synthesis) were assembled and amplified with PCR. The bands are clear, which is good for gel recycling. The concentrations of gel recycled blocks were all over 150 ng/ul, far higher than those in method (A). The result of lane 4 indicates that the blocks were capable of joining to the full-length gene with PCR.

After gel recycling the target gene bands in Fig. 5 and Fig. 6, we ligated them to T-vectors, respectively, and sent them for sequencing verification. The result of sequence alignments indicated that the target sequence in the T-vector was the same as 'beta_optimized.fasta',

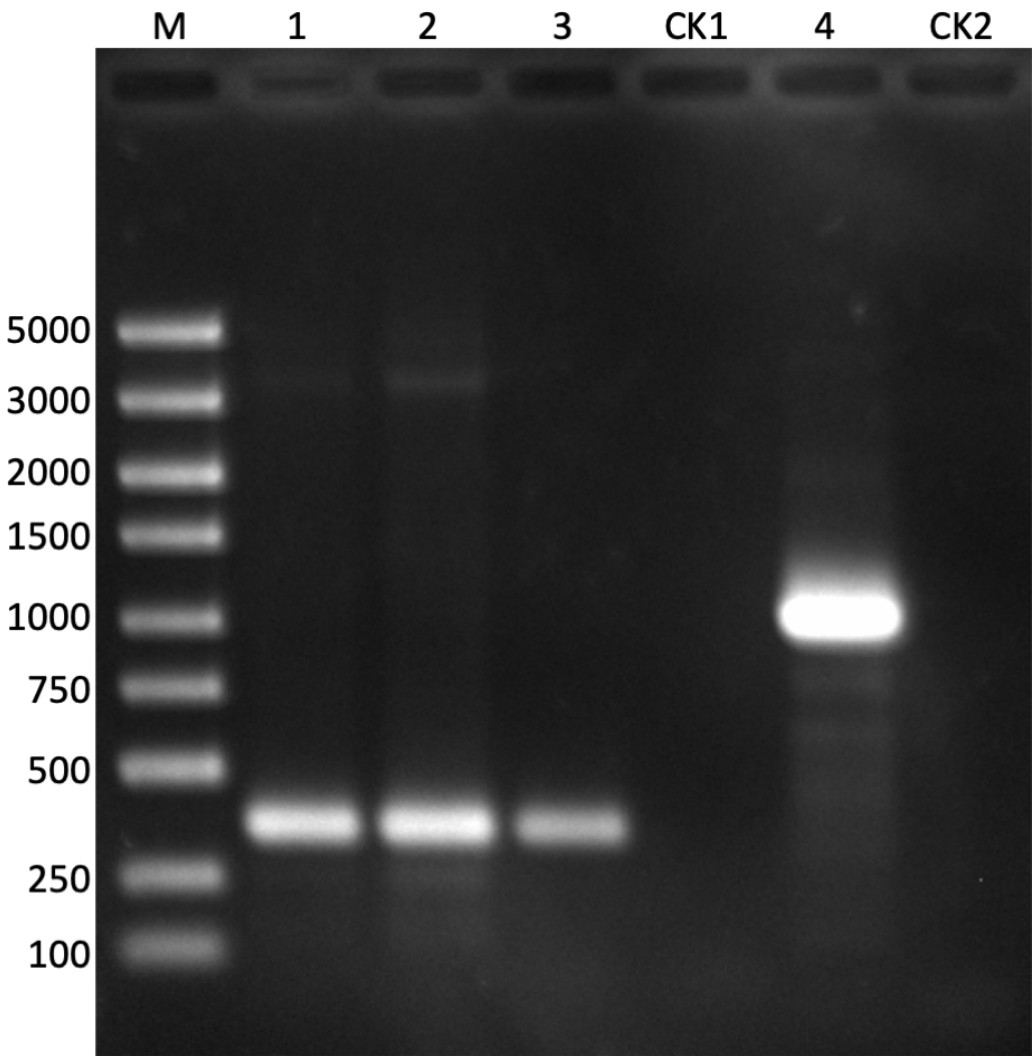

**Figure 6** **The validation of method (B).** M: DNA marker; lane 1: block1; lane 2: block2; lane 3: block3; CK1: negative control for lanes 1–3 with ddw as the template; lane 4: blocks 1–3; CK2: negative control for lane 4 with ddw as the template.

for both method (A) and method (B) (https://github.com/shiqiang-lin/gene-synthesis). These results demonstrated that the fragments by method (A) and method (B) with our program could be used to assemble the full-length gene with PCR.

## DISCUSSION

In this study, we design the algorithm and write the Python code according to the principle of gene synthesis with fragment assembly *via* PCR. The program has been experimentally verified with the example gene.

Compared with other gene design programs, our program has several advantages. The program is written with Python, the most popular programming language nowadays. The code can be read easily, which facilitates the understanding and adjustment of the program
if necessary. We are now in a time of artificial intelligence and there are more and more people learning and using Python. Thus, our program is in a friendly environment for lab application. We provide detailed algorithm description, source code, and annotation. These resources make possible modification and improvement of the program to meet with the specific needs of more users. The fragment length recommended in the program is 59, however, users may decide this value themselves. The users only need to change the source code in line 192 with IDLE, for example, from "set_fragment_len = 59" to "set_fragment_len = 69". This enables users to control the quality and cost of the commercial synthesis of fragments. Normally, the base price is more expensive for longer fragments. It is necessary to set a reasonable fragment length for cost-effective commercial synthesis. In addition, it is possible to change the screening range of PCR annealing temperatures, which is done by modifying the source code in line 210 with IDLE, *i.e.,* "for Tm_set in range(54,69):". In this situation, the PCR annealing temperatures screened are 54, 55, 56, ..., 68. If we change (54,69) to (57,63), then the PCR annealing temperatures screened will be from 57 to 63, with 63 excluded.

Moreover, the program makes it possible to flexibly divide the full-length gene into blocks. Previous study has shown that up to seven overlapped DNA fragments can be assembled *via* PCR (*Xiong et al., 2004*). Thus, for example, if we have 20 DNA fragments to be joined, then three blocks containing six, seven, and seven fragments respectively may be defined. In this situation, block1 (fragments 1–6) is overlapped with block2 (fragments 7–13) in that fragment 6 is overlapped with fragment 7, and block2 (fragments 7–13) is overlapped with block3 (fragments 14–20) as well. In the case of 22 DNA fragments, four blocks containing five, five, six, and six fragments respectively may be defined, and so on. In this way, each block has several DNA fragments, with the 5′ end of the first fragment overlapping the 3′ end of the last fragment of the previous block, and the 3′ end of the last fragment overlapping the 5′ end of the first fragment of the next block.

As the program lists the overlap sequences and the gene primers, all with close Tm, the primers for each block can be easily selected. For example, to amplify block1, the sense_gene_primer (in 'gene_primers.txt') is used as the forward primer, and the reverse complement sequence of the overlap sequence (the second column of 'all_sense_fragments.txt' for Method A and the second column of 'sense_antisense_fragments.txt' for Method B) of the last fragment in block1 is used as the reverse primer. To amplify block2, the user can use the overlap sequence of the last fragment of block1 as forward primer and the reverse complement sequence of the last fragment in block2 as reverse primer, and so on. It is worth mentioning that in the 'all_sense_fragments.txt' and the 'sense_antisense_fragments.txt', the program lists the overlap sequences in the second column with sense sequences. Therefore, to get the reverse primer for a block, the user needs to perform reverse complementation to the overlap sequence of the last fragment of the block.

Furthermore, our program provides users with two methods of fragment assembly, which are method (A) and method (B), to increase the flexibility of the experimental process. Under some circumstances, users may make adjustments based on methods (A) and (B). Any one or more fragments can be changed to its or their reverse complement

sequences (while block primers are unchanged). This offers users even more possible methods of assembling fragments.

## CONCLUSIONS

This study solves the problem of gene synthesis design with the Python programming language. We provide a user-friendly, free, open-source, and experimentally validated method for the design of DNA fragments for PCR gene synthesis. Our method will lend support to the study of protein structure and function, protein engineering, synthetic biology, and so on.

### Funding

This study is supported by the Spark Project of the Fujian Provincial Department of Science and Technology (NO. 2023S0012). The funders had no role in study design, data collection and analysis, decision to publish, or preparation of the manuscript.

### Grant Disclosures

The following grant information was disclosed by the authors:
The Spark Project of the Fujian Provincial Department of Science and Technology: NO. 2023S0012.

### Competing Interests

The authors declare there are no competing interests.

### Author Contributions

- Yunzhuo Hu performed the experiments, analyzed the data, prepared figures and/or tables, authored or reviewed drafts of the article, and approved the final draft.
- Danni Pan performed the experiments, prepared figures and/or tables, and approved the final draft.
- Fei Xu performed the experiments, prepared figures and/or tables, and approved the final draft.
- Bifang Huang performed the experiments, prepared figures and/or tables, and approved the final draft.
- Xuanyang Chen conceived and designed the experiments, analyzed the data, authored or reviewed drafts of the article, and approved the final draft.
- Shiqiang Lin conceived and designed the experiments, performed the experiments, analyzed the data, prepared figures and/or tables, authored or reviewed drafts of the article, and approved the final draft.

### Data Availability

The program source code, the example gene sequence file, the output files, and the resulting files of experimental validation, are available at GitHub and Zenodo:

- https://github.com/shiqiang-lin/gene-synthesis.
- shiqiang-lin. (2024). shiqiang-lin/gene-synthesis: gene_synthesis (v0.1). Zenodo. https://doi.org/10.5281/zenodo.12560364.

The script along with the documentation is also available at PyPI at https://pypi.org/project/gene-synthesis/0.1/ for a convenient installation with Python pip.

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
