# Peer review of "Gene synthesis design: a pythonic approach"

_PeerJ, doi:10.7717/peerj.17750_

## Round 0.1 · original submission · Major Revisions

Please consider all the comments and submit a revised version at your convenience, along with a rebuttal letter.

Reviewer 1 ·

Basic reporting

The paper is of general interest and is overall very well written. It will be helpful if the author could add brief instructions on installation of this script for all systems (Mac, Windows and Linux) for the convenience of the users. For example, I personally don't have a Mac and I installed python3.10 on Window10, then I used VS code terminal installed matplotlib and biopython with pip install. It took me sometime to figure out, but it might takes longer for other user who's not familiar with python environment in a different system.

Experimental design

No comment

Validity of the findings

For large genes, similarly as the author illustrated example with gene lengths longer than 1k, one needs to merge blocks of nine fragments first, and then ligate the PCR product, it will be helpful if the author could suggest on choosing or designing the primers to amplify each of the gene block.

Additional comments

In general this Python script would be very useful for researchers on Synthetic Biology and routine gene cloning. Recommend to publish if minor issues are answered and addressed.

Cite this review as

Reviewer 2 ·

Basic reporting

The authors report new software to design fragments for PCR assembly of DNA. I do agree with the authors that this is a useful cloning technique and that software support will help many applications, in particular in protein engineering. However I believe the paper has quality problems that would prevent its success with the community.

I don't think the state of the art is being represented fairly. The authors recognize that previous work exists but brush this off by saying previous tools are "old-fashioned" and "not written in python" which feels very hand-wavy.

The level of english could be improved throughout the article, the wrong words are applied at many places (delicate, pictorial). In most cases the level of writing doesn't prevent the understanding, but I found that it was a real problem in the description of the algorithm. Sentences like "we achieve algorithm design and code implementation according to the principle of gene synthesis (...)" should be checked for grammatical correctness.

The tool itself is simply a python file in a Github repository without any README, no general documentation, and non-standard in-code documentation (for some reason docstrings are outside the python functions) which won't help users adopt the tool or build on top of it. Moreover, the tool is only usable from the command line, and so it doesn't matter that much to the user whether it was written in python or another language. If the tool was better documented and installable as a python library it would be much more useful to the community.

Figure legends are difficult to understand for who wouldn't be familiar with the field. They have the minimum of explanations, and use vocabulary like "merging block" which is not explained in the paper.

Experimental design

Non applicable (the authors are presenting a new tool, not testing an hypothesis). The experimental validation of the fragments returned by the software is nice and indeed a reassurance that the software output is valid.

Validity of the findings

non-applicable, the authors do not claim findings, only a new tool

Additional comments

None

Cite this review as

---

## Round 0.2 · accepted · Accept

Thanks for addressing all the reviewers' concerns; your paper is now accepted by PeerJ.

Reviewer 1 ·

Basic reporting

No comment

Experimental design

No comment

Validity of the findings

No comment

Additional comments

The author properly addressed and answered all questions raised by reviewers. Recommend to publish.

Cite this review as